# Towards Adversarial Robustness of Bayesian Neural Network through Hierarchical Variational Inference

## Abstract

Recent works have applied Bayesian Neural Network (BNN) to adversarial training, and shown the improvement of adversarial robustness via the BNN's strength of stochastic gradient defense. However, we have found that in general, the BNN loses its stochasticity after its training with the BNN's posterior. As a result, the lack of the stochasticity leads to weak regularization effect to the BNN, which increases KL divergence in ELBO from variational inference. In this paper, we propose an enhanced Bayesian regularizer through hierarchical variational inference in order to boost adversarial robustness against gradient-based attack. Furthermore, we also prove that the proposed method allows the BNN's stochasticity to be elevated with the reduced KL divergence. Exhaustive experiment results demonstrate the effectiveness of the proposed method by showing the improvement of adversarial robustness, compared with adversarial training (Madry et al., 2018) and adversarial-BNN (Liu et al., 2019) under PGD attack and EOT-PGD attack to the $L_\infty$ perturbation on CIFAR-10/100, STL-10, and Tiny-ImageNet.

## 1 Introduction

Deep neural networks have achieved impressive performance in a wide variety of machine learning tasks. Despite the breakthrough outcomes, deep neural networks are easily deceived from adversarial attack with the carefully crafted perturbations (Szegedy et al., 2014; Goodfellow et al., 2015; Chen et al., 2017; Carlini & Wagner, 2017; Papernot et al., 2017; Eykholt et al., 2018; Madry et al., 2018). Injecting these perturbations into clean inputs (i.e., adversarial examples), which are imperceptible to the human eyes, fools the estimators in the deep neural networks. Weak reliability due to the invisible perturbations has affected security problems in deep learning applications (Apruzzese et al., 2019; Wang et al., 2019b; Sagduyu et al., 2019; Rosenberg et al., 2020).

To defend such adversarial examples, many algorithms have been studied to improve adversarial robustness so far. *Adversarial training*, where deep neural networks are trained on adversarial examples, is one of the few defense strategies against strong adversarial attacks (Huang et al., 2015; Zantedeschi et al., 2017; Kurakin et al., 2017; Madry et al., 2018; Athalye et al., 2018a; Liu et al., 2018). Among them, Madry et al. (2018) has shown that adversarially trained networks can be robust to white-box attacks with the knowledge of the network parameters. Besides, most of the above studies have agreed with that adversarial training shows an effective adversarial robustness against several white-box attacks.

Meanwhile, adversarial training and BNN have been combined to improve adversarial robustness with stochastic approach through variational inference. In fact, the variational inference maximizes Evidence Lower Bound (ELBO) to find an approximate posterior closely following the true posterior for the machine learning tasks (Graves, 2011; Kingma & Welling, 2014; Blundell et al., 2015; Hernández-Lobato & Adams, 2015). Based on the variational inference, adversarial training with BNN has accomplished the achievement of adversarial robustness by implicitly using the approximate posterior against the adversarial perturbations (Mescheder et al., 2017; Ye & Zhu, 2018). They have focused on training the network parameters or spaces itself on the maximum ELBO without obtaining the approximate posterior directly.

Contrary to the above studies, Liu et al. (2019) presents an adversarial training with BNN, called "adversarial-BNN" to deal with the approximate posterior explicitly. They straightforwardly learn Gaussian parameters (e.g., mean and variance) of the approximate posterior as follows: $w \sim \mathcal{N}(\mu, \sigma^2)$, instead of the weight parameters. Alternatively, the weight parameters are sampled by the approximate posterior, such that $w = \mu + \sigma\epsilon$, where the stochastic sampler $\epsilon$ is gererated from $\epsilon \sim \mathcal{N}(0, 1)$. The stochastic sampler $\epsilon$ provides the change of the weight parameters with the learned Gaussian parameters. The variation of them creates stochastic gradient, helping the improvement of adversarial robustness (Carbone et al., 2020).

However, we find that in general, the approximate posterior's variance converges to zero-like small value as follows: $w \sim \mathcal{N}(\mu, \sigma^2 \approx 0)$ after training the BNN with its posterior. Although the stochastic sampler helps the weight parameters to change, they become fixed-like parameters, such that $w = \mu + \sigma(\approx 0)\epsilon$. The lack of their stochasticity causes the BNN's stochasticity to be vanished so that the BNN cannot easily respond to slightly different inputs within the same class. In other words, the vanished stochasticity breaks the regularization effect in the BNN, which increases KL divergence in the ELBO. The broken BNN regularizer produces an ill-posed posterior, thus resulting in weak adversarial robustness. This is because the broken regularizer hinders the maximum ELBO from approximating the true posterior against the adversarial perturbations. Therefore, an enhanced BNN regularizer is required to better approximate the true posterior for adversarial robustness.

In this paper, we present the enhanced Bayesian regularizer through hierarchical variational inference in order to boost adversarial robustness compared to the BNN regularizer from variational inference. Furthermore, we also prove that the proposed method significantly intensifies the BNN's stochasticity by introducing a closed form approximation of conjugate prior for the true posterior. In the end, we validate the effectiveness of the proposed method by showing the improvement of adversarial robustness, compared with adversarial training (Madry et al., 2018) and adversarial-BNN (Liu et al., 2019) under PGD attack as well as EOT-PGD attack to the $L_\infty$ perturbation on CIFAR-10/100, STL-10, and Tiny-ImageNet.

Our contributions of this paper can be summarized into two-fold as follows.

- We newly design an enhanced Bayesian regularizer through hierarchical variational inference built with a concept of the conjugate prior, and verify that the proposed method further strengthens the BNN's stochasticity, compared to the BNN regularizer based on variational inference.
- We conduct exhaustive experiments to validate the effectiveness of the proposed method by adversarial robustness, and exhibit the outstanding performance compared with adversarial training and adversarial-BNN under both PGD attack and EOT-PGD attack on four benchmark datasets: CIFAR-10/100, STL-10, and Tiny-ImageNet.

## 2 PRELIMINARY

In this section, we specify the notations used in our paper at first and summarize the related works on adversarial attack/defense, and adversarial training with BNN.

**Notations.** Let $x$ denote the clean image from a given dataset, and $y$ denote the class label corresponding to the clean image. Let $\mathcal{D}$ and $\mathcal{D}^{adv}$ indicate each clean dataset and adversarial dataset, such that $(x, y) \sim \mathcal{D}$ and $(x^{adv}, y) \sim \mathcal{D}^{adv}$. A deep neural network $f$ parameterized by weight parameters $w$ is denoted by $f_w(x)$. Adversarial examples are represented by $x^{adv} = x + \delta$, where $\delta$ denotes the adversarial perturbations. In order to align the experiments in the previous works, we use the cross-entropy loss $J(f_w(x), y)$ for image classification. Moreover, we regard $\delta$ as the $L_\infty$ perturbation within $\gamma$-ball, such that $\|\delta\|_\infty \leq \gamma$. Here, $\|\cdot\|_\infty$ describes the $L_\infty$.

### 2.1 ADVERSARIAL ATTACK/DEFENSE

**Adversarial Attacks.** The goal of adversarial attacks is generating the adversarial examples to deceive the prediction of the deep neural networks. Most of them produce the adversarial examples by the gradient of the loss function over the input. Goodfellow et al. (2015) introduces a single-step attack called Fast Gradient Sign Method (FGSM). Kurakin et al. (2017) proposes iterative-FGSM with multiple-step attack. Further, Carlini & Wagner (2017) presents C&W attack to overcome

defensive distillation (Papernot et al., 2016), encompassing a range of attacks cast from the same optimization framework. On the other hand, Athalye et al. (2018a) analyzes the effectiveness of Projected Gradient Descent (PGD) method (Madry et al., 2018) to perform adversarial attack to the $L_\infty$ perturbation. PGD attack is an iterative algorithm by computing the gradient of the loss in the direction of the highest loss and projecting it back to the $L_\infty$ perturbation around the clean image. We use the PGD attack for the experiments of adversarial robustness, which can be written as:

$$x_{t+1}^{adv} = \prod_{x,\gamma} \left[ x_t^{adv} + \eta \cdot \text{sign} \left( \nabla_x J \left( f_w(x), y \right) \big|_{x=x_t^{adv}} \right) \right], \tag{1}$$

where $\prod_{x,\gamma}$ is the function to project its argument to the surface of $x$'s $\gamma$-neighbor ball $\left\{ x \big| \left\| x^{adv} - x \right\|_\infty \leq \gamma \right\}$, and $\eta$ denotes step size. For the adversarial defense utilizing randomized classifiers, an adaptive attack such as Expectation over Transformation (EOT) is known to be effective because it allows an attacker to compute the actual gradient over the expected transformation to the stochastic classifiers (Athalye et al., 2018a;b). Inspired from the EOT attack, we can modify Eq. (1) and adjust more efficient PGD attack (Zimmermann, 2019).

$$x_{t+1}^{adv} = \prod_{x,\gamma} \left[ x_t^{adv} + \eta \cdot \text{sign} \left( \mathbb{E}_w \left[ \nabla_x J \left( f_w(x), y \right) \big|_{x=x_t^{adv}} \right] \right) \right], \tag{2}$$

Zimmermann (2019) applies the averaged-gradient over the multiple weight parameters to generate the adversarial examples, and shows that this efficient PGD attack degrades the BNN's adversarial robustness. This is because the averaged-gradient can attack all of the possibly sampled weight parameters in the BNN. In other words, the averaged-gradient attack weakens the BNN's advantage of stochastic gradient defense. In this paper, we call the efficient PGD attack, namely *EOT-PGD attack*, and take it to validate the effectiveness of the proposed method.

**Adversarial Defense.** The aim of adversarial defense is to secure deep learning networks against adversarial attacks. There are generally three main categories of defense strategies (Silva & Najafi-rad, 2020; Hao-Chen et al., 2020). (1) Gradient masking/obfuscation: a defender intentionally hides the gradient information of deep neural networks in order to confuse adversaries, because most attack algorithms are based on the classifier's gradient information (Papernot et al., 2016; Buckman et al., 2018; Guo et al., 2018; Dhillon et al., 2018). (2) Adversarial detections: they focus on distinguishing whether the input is benign or adversarial (Metzen et al., 2017; Grosse et al., 2017; Xu et al., 2017). When they succeed to detect the adversarial attack in the input, the classifiers stop making a decision. (3) Robust optimization: it is a well-known paradigm that aims to obtain solutions under bounded feasible regions. Especially, its main interest from an adversarial perspective, is improving the classifier's robustness by changing the learning scheme of deep neural networks (Cisse et al., 2017; Hein & Andriushchenko, 2017; Raghunathan et al., 2018; Wong & Kolter, 2018).

Adversarial training (Goodfellow et al., 2015; Kurakin et al., 2017; Madry et al., 2018) is one of the famous robust optimization methods in deep learning fields, which allows the deep neural network to learn robust parameters against the adversarial examples. Madry et al.(2018) tries to find the optimal weight parameters against adversarial dataset $\mathcal{D}^{adv}$ to improve adversarial robustness under PGD attack to the $L_\infty$ perturbation, by minimizing the pre-defined loss $\mathcal{L}$ for the machine learning tasks. The formulation can be written as:

$$w^* = \arg\min_w \mathbb{E}_{(x^{adv},y) \sim \mathcal{D}^{adv}} \left[ \mathcal{L} \left( f_w(x^{adv}), y \right) \right], \tag{3}$$

where $x^{adv}$ is generated from the PGD attack as described in Eq. (1). The optimal weight parameters $w^*$ get the capacity of adversarial defense to confront the adversarial perturbations.

## 2.2 Adversarial training with BNN

Many studies have mingled with adversarial training and BNN through variational inference. Here, variational inference imposes the probabilistic approach on the network parameters or spaces to obtain a closed form approximation of the true posterior based on the maximum ELBO, thus leading to well-posed inference (Welling & Teh, 2011; Paisley et al., 2012; Hoffman et al., 2013; Salimans & Knowles, 2013; Kingma & Welling, 2014; Rezende et al., 2014). Likewise, finding the closed form from an adversarial perspective can lead to well-posed inference for adversarial robustness. Thus,

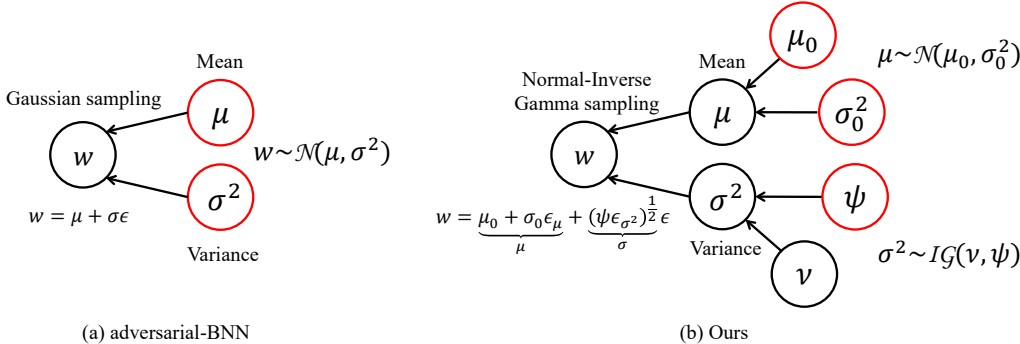

Figure 1: Diagrams that describe the methods of sampling the weight parameters for (a) with adversarial-BNN and (b) the proposed method. The weight parameters in (a) are sampled from the Gaussian distribution $q(w)$ as an approximate posterior through variational inference, and those in (b) are sampled from the Normal-Inverse Gamma distribution $q(\mu, \sigma^2)$ as an approximate conjugate prior through hierarchical variational inference. Red circles denote the learning parameters in the BNN, black circles denote the random variables sampled from the described probability distributions, and the black arrows indicate the sampling order.

adversarial training with BNN tries to approximate a true posterior against adversarial perturbations: $p(w \mid \mathcal{D}^{adv})$ to indirectly utilize it to conduct the well-posed inference for adversarial robustness. Fundamentally, the reason we should approximate the true posterior is attributed by its intractability because of a high dimensional dependency of adversarial dataset $p(\mathcal{D}^{adv})$. Thus, variational inference for adversarial robustness introduces an approximate posterior in the closed form of $q(w)$ as a practical estimator to approximate the true posterior, where the formulation can be written as:

$$
\begin{aligned}
\log p(y \mid x^{adv}) = \mathcal{D}_{KL}(q(w) \,||\, p(w \mid \mathcal{D}^{adv})) \\
+ \underbrace{\mathop{\mathbb{E}}_{w \sim q(w)} \left[ \log p(y \mid x^{adv}, w) \right] - \mathcal{D}_{KL}(q(w) \,||\, p(w))}_{\text{ELBO}}.
\end{aligned}
\tag{4}
$$

We should minimize KL divergence $\mathcal{D}_{KL}\left(q(w) \,||\, p(w \mid \mathcal{D}^{adv})\right)$ above to make the probability-distance close between the two probability distributions. But, this KL divergence is still intractable measurement, and we can alternatively maximize the ELBO in Eq. (4), which has the same effect to minimize this KL divergence. This is because the log-probability $\log p(y \mid x^{adv})$ in Eq. (4) is consistent with the change of the weight parameters $w$. The formulation of variational inference also can be represented to lower bound as: $\log p(y \mid x^{adv}) \geq \textbf{ELBO}$.

Some of the studies (Mescheder et al., 2017; Ye & Zhu, 2018; Wang et al., 2019a; Carbone et al., 2020) are to improve adversarial robustness by maximizing the ELBO. Others (Feinman et al., 2017; Smith & Gal, 2018; Wang et al., 2018) are to detect the adversarial examples by predicting Bayesian uncertainty (Gal & Ghahramani, 2016; Kendall & Gal, 2017; Pearce et al., 2020), which are also based on the maximum ELBO. These previous studies have implicitly examined the approximate posterior to better learn the network parameters or spaces, instead of directly shaping the approximate posterior.

Unlike the previous studies, Liu et al. (2019) proposes "adversarial-BNN" which combines adversarial training and the BNN to improve adversarial robustness by shaping the approximate posterior explicitly. Liu et al. designs its posterior $q(w) = \mathcal{N}(\mu, \sigma^2)$ and considers Gaussian parameters (e.g., mean, variance) of it as learning parameters. In this work, the first term in the ELBO represents the expected log-likelihood calculated from the cross-entropy as follows: $\frac{1}{N} \sum_{i=1}^{N} J\left(f_{w^{(i)}}(x^{adv}), y\right)$ by Monte-Carlo sampling, where $N$ denotes the sampling number. As shown in Fig. 1(a), the weight parameter $w^{(i)}$ is sampled by the approximate posterior's mean and variance $\mu, \sigma^2$, such that $w^{(i)} = \mu + \sigma \epsilon^{(i)}$. Here, $\epsilon^{(i)}$ is the stochastic sampler, such that $\epsilon^{(i)} \sim \mathcal{N}(0, 1)$.

The second term in the ELBO normally represents a regularization term for an approximate posterior as the BNN's posterior. This term can be computed from the KL divergence between the approximate posterior $q(w)$ and a prior $p(w)$ which is usually set to $\mathcal{N}(0, 1)$ for variational inference. In other words, this KL divergence has a role of a BNN regularizer in ELBO, and this regularizer is an

important factor for adversarial robustness. This is because once the BNN regularizer is increased unexpectedly, the minimum ELBO produces an ill-posed posterior, creating weak adversarial robustness. In contrast, as the BNN regularizer is reduced, the maximum ELBO allows for the well-posed inference due to strong regularization effect bringing in better approximating the true posterior. In addition to the aspect of the ELBO, there are many studies showing that high regularization performance improves adversarial robustness (Saito et al., 2018; Yu et al., 2019; Zhang & Liang, 2019; Sun et al., 2019; Terjék, 2020; Zhang et al., 2020).

## 3 PROPOSED METHOD

Adversarial training with BNN has expanded generic probability to adversarial probability, and has shown the improvement of adversarial robustness. However, as mentioned in the Section 1, the vanished stochasticity declines adversarial defense functionality of the BNN. To investigate the BNN's stochasticity, we introduce a concept of the conjugate prior, and develop a new ELBO motivated by the conjugate prior. The conjugate prior can provide a key to how the Gaussian parameters of the approximate posterior $q(w)$ are trained during adversarial training. Based on our findings in the conjugate view, we will design an enhanced Bayesian regularizer that solves the stochasticity problem causing weak adversarial robustness. Before we explain the proposed method, we firstly introduce two parts: (1) the conjugate prior of the Gaussian posterior, (2) hierarchical variational inference for building a new ELBO with this concept of conjugate prior.

**Gaussian Conjugate Prior.**  The conjugate prior of the Gaussian posterior is called the Gaussian conjugate prior. In the conjugate view, the true posterior $p(w \mid \mathcal{D})$ is no longer posterior, but is the likelihood $p(w \mid \mathcal{D}, \mu, \sigma^2)$ given the Gaussian parameters $\mu, \sigma^2$ when assuming the true posterior is Gaussian. Here, Gaussian conjugate prior is the probability distribution $p(\mu, \sigma^2 \mid \mathcal{D})$ for the Gaussian parameters of the true posterior. The conjugate prior takes an benefit to approximate the true posterior by modeling such Gaussian parameters, thus guiding well-posed posterior. But, the conjugate prior is also intractable to deal with, similar to the true posterior because they are same distribution family. Therefore, we introduce an approximate conjugate prior $q(\mu, \sigma^2)$, in the shape of Normal-Inverse Gamma distribution, to approximate the conjugate prior of the true posterior. The formulation of the approximate conjugate prior can be written as follows:

$$q(\mu, \sigma^2) = \mathcal{N}\left(\mu \mid \mu_0, \sigma_0^2\right) \mathcal{IG}\left(\sigma^2 \mid \nu, \psi\right). \tag{5}$$

It can be factorized into the Normal distribution $q(\mu) = \mathcal{N}\left(\mu \mid \mu_0, \sigma_0^2\right)$ and the Inverse Gamma distribution $q(\sigma^2) = \mathcal{IG}\left(\sigma^2 \mid \nu, \psi\right)$ independently by mean-field approximation (Kingma & Welling, 2014). In the Normal distribution, $\mu_0$ denotes the mean prior knowledge of the approximate posterior $q(w)$, and $\sigma_0^2$ denotes the variance of the approximate posterior's mean. In addition, $\nu$, in the Inverse Gamma distribution, denotes the degree of freedom on positive real set which determines the shape of this distribution, and $\psi$ denotes the variance prior knowledge of the approximate posterior.

Here, the approximate conjugate prior can sample each of the Gaussian parameters independently. The Normal distribution samples the approximate posterior's mean, and the Inverse Gamma distribution samples its variance. As illustrated in Fig. 1(b), tractable parameters $\mu_0, \sigma_0^2, \psi$, and $\nu$ in $q(\mu, \sigma^2)$ can generate the mean $\mu$ and the variance $\sigma^2$ of the approximate posterior, where $\mu = \mu_0 + \sigma_0 \epsilon_\mu$, and $\sigma^2 = \psi \epsilon_{\sigma^2}$, such that $\epsilon_\mu \sim \mathcal{N}(0, 1)$, $\epsilon_{\sigma^2} \sim \mathcal{IG}(\nu, 1)$. In addition, the generated mean and the variance are both used to sample the weight parameters: $w = \mu_0 + \sigma_0 \epsilon_\mu + (\psi \epsilon_{\sigma^2})^{1/2} \epsilon$. We call it Normal-Inverse Gamma sampling. More importantly, the approximate conjugate prior has more number of the tractable parameters than the approximate posterior $q(w)$, which significantly help sampling the weight parameters for complex representation.

**Hierarchical Variational Inference.**  Next, we tackle the way the approximate conjugate prior makes a closed form approximation of the conjugate prior as a same family of true posterior. We modify the ELBO in Eq. (4) to build up a new ELBO with the approximate conjugate prior, namely *hierarchical-ELBO*, which can be written as:

$$\mathbb{E}_{(\mu, \sigma^2) \sim q(\mu, \sigma^2)} \left[\log p(y \mid x^{adv}, \mu + \sigma \epsilon)\right] - \mathcal{D}_{KL}(q(\mu, \sigma^2) \mid\mid p(\mu, \sigma^2)). \tag{6}$$

The first term stands for the classification performance taken from the cross-entropy. The second term represents the BNN regularizer in hierarchical-ELBO equivalent to that in the ELBO as

depicted in Eq. (4). The second term can be computed by the KL divergence between the approximate conjugate prior $q(\mu, \sigma^2)$ and a hierarchical prior $p(\mu, \sigma^2)$. Note that the hierarchical prior is factorized into the mean prior $p(\mu \mid \sigma^2)$ and the variance prior $p(\sigma^2)$. Then, the above KL divergence can be decomposed into two equations: (1) $\mathbb{E}_{\sigma^2 \sim q(\sigma^2)} \left[ \mathcal{D}_{KL}(q(\mu) \mid\mid p(\mu \mid \sigma^2)) \right]$ and (2) $\mathcal{D}_{KL}(q(\sigma^2) \mid\mid p(\sigma^2))$ (see Appendix A). In order to make two equations tractable, both the mean prior and the variance prior should be set, similar to the setting of the prior $p(w) = \mathcal{N}(0, 1)$ in the ELBO from variational inference. Since the prior is Gaussian, the hierarchical prior can be formed to Gaussian conjugate prior, Normal-Inverse Gamma, where we keep the prior knowledge of the mean and the variance with zero and one, respectively. We set the mean prior to $\mathcal{N}\left(0, \lambda^{-1}\sigma^2\right)$ for (1), where $\lambda$ denotes inverse scale factor to the variance $\sigma^2$. In addition, the variance prior is set to $\mathcal{IG}(\nu, 1)$ for (2). Here, $\nu$ is set to the same parameters both in $q(\sigma^2)$ and $p(\sigma^2)$ to reduce computational complexity in calculating (2).

On the above settings for the hierarchical prior $p(\mu, \sigma^2)$, the second term can be tractably expanded with the summation of the two equations (see Appendix B), which can be written as:

$$\mathcal{D}_{KL}(q(\mu, \sigma^2) \mid\mid p(\mu, \sigma^2)) = \frac{1}{M} \sum_{j=1}^{M} [\nu_j \frac{\lambda_j(\sigma_{0,j}^2 + \mu_{0,j}^2) + 2}{2\psi_j} + \log \frac{\psi_j^{\frac{2\nu_j + 1}{2}}}{(\lambda_i \sigma_{0,j}^2)^{\frac{1}{2}}} - \frac{2\nu_j + 1}{2} - \frac{1}{2}G(\nu_j)], \quad (7)$$

where $M$ denotes the number of weight parameters in the deep neural networks, and $j$ represents the $j^{th}$ parameters for the $j^{th}$ approximate conjugate prior. The function $G$ has the formulation of Digamma function, such that $G(\nu_j) = \Gamma'(\nu_j)/\Gamma(\nu_j)$.

**ELBO vs Hierarchical-ELBO.** Until we control the parameters $\lambda$, $\nu$ of the hierarchical prior $p(\mu, \sigma^2)$, the ELBO is essentially same to the hierarchical-ELBO. Also, the BNN regularizer in the ELBO is identical to that in the hierarchical-ELBO. This is because it is nothing but we just have replaced the weight parameters in the ELBO with the Gaussian parameters for the extension, as described in Eq. (6). Different from the fixed prior $p(w) = \mathcal{N}(0, 1)$ in the ELBO, controllable parameters $\lambda$, $\nu$ of the hierarchical prior $p(\mu, \sigma^2)$ can allow the BNN regularizer in the hierarchical-ELBO to change adaptively. As a consequence, properly chosen parameters $\lambda$ and $\nu$ can reduce the KL divergence, helping to well-posed posterior with the maximum hierarchical-ELBO. The formulation of the hierarchical-ELBO can be also represented to lower bound with the ELBO as follows: $\log p(y \mid x^{adv}) \geq$ **hierarchical-ELBO** $\geq$ **ELBO**, where the hierarchical-ELBO better approximates the true posterior than the ELBO.

**Enhanced Bayesian Regularizer.** Now, we focus on the optimization of the BNN regularizer in the hierarchical-ELBO to reduce the KL divergence. By gradient results over the controllable parameters: $\frac{\partial}{\partial \lambda_j} \mathcal{D}_{KL} = 0$ and $\frac{\partial}{\partial \nu_j} \mathcal{D}_{KL} = 0$, we can easily obtain the optimal value $\lambda^* = \frac{\psi}{\nu(\sigma_0^2 + \mu_0^2)}$, but the optimal value $\nu^*$ is intractable to acquire, since the gradient result is intractable as follows: $\frac{1}{2}\frac{d}{d\nu}G(\nu) - \frac{1}{2\nu} = \frac{1}{\psi} + \log\psi - 1$. We should indirectly find the optimal value $\nu^*$ by applying the above gradient results to the KL divergence, which can be written as:

$$\mathcal{D}_{KL}(q(\mu, \sigma^2) \mid\mid p(\mu, \sigma^2)) = \frac{1}{M} \sum_{j=1}^{M} [\frac{1}{2}\log\left(1 + \frac{\mu_{0,j}^2}{\sigma_{0,j}^2}\right) + \underbrace{\frac{\nu_j}{2}\frac{d}{d\nu_j}G(\nu_j) - \frac{1}{2}G(\nu_j) + \frac{1}{2}\log\nu_j - \frac{1}{2}}_{H(\nu_j)}], \quad (8)$$

where the function $H(\nu_j)$ is decreasing and convergent function (see Appendix C). We examine this function is converged to zero, as $\nu_j$ gets large enough. With the optimal value $\nu^*$ infinite, the KL divergence in the hierarchical-ELBO can be definitely reduced.

Finally, we propose the enhanced Bayesian regularizer as follows: $\frac{1}{M} \sum_{j=1}^{M} [\frac{1}{2}\log\left(1 + \frac{\mu_{0,j}^2}{\sigma_{0,j}^2}\right)]$. The proposed method provides more tight lower bound than the ELBO, such that the formulation can be described as follows: $\log p(y \mid x^{adv}) \geq$ **hierarchical-ELBO** $\gg$ **ELBO**. Based on the tight lower bound, the enhanced Bayesian regularizer aids to the acquisition of the well-posed posterior to boost adversarial robustness. Consequently, we have verified that the proposed method considerably intensifies the BNN regularizer by designing the hierarchical-ELBO in the conjugate view. In practical, we can compute the proposed method for each training iteration, such as Expectation-Maximization (EM) step. We describe the algorithm for the proposed method below.

---

**Algorithm 1** Code for adversarial training in the BNN with the enhanced Bayesian regularizer

---

1: **Require:** $\alpha$ (learning rate), $\mu_0$ and $\sigma_0$ (learning parameters)
2: **for** $(x, y)$ in $\mathcal{D}$ **do**                                          ▷ Training procedure per epoch
3:     $w \leftarrow \mu_0 + \sigma_0 \epsilon, \epsilon \sim \mathcal{N}(0, 1)$             ▷ Sample the weight parameters
4:     $x^{adv} \leftarrow \text{attack}(x, y, w)$                    ▷ PGD attack to generate adversarial examples
5:     $\hat{y} \leftarrow f_w(x^{adv})$                ▷ Predict the outputs given the adversarial examples
6:     $\mathcal{L}_1 \leftarrow J(\hat{y}, y)$                              ▷ Calculate the cross-entropy loss
7:     $\mathcal{L}_2 \leftarrow \sum_{j=1}^{M} \frac{1}{2} \log(1 + \frac{\mu_{0,j}^2}{\sigma_{0,j}^2})$                ▷ Calculate the proposed method
8:     $\mathcal{L} \leftarrow \mathcal{L}_1 + \mathcal{L}_2$                    ▷ Calculate the hierarchical-ELBO in Eq. (6)
9:     $(\mu_0, \sigma_0) \leftarrow (\mu_0 - \alpha \frac{\partial \mathcal{L}}{\partial \mu_0}, \sigma_0 - \alpha \frac{\partial \mathcal{L}}{\partial \sigma_0})$                ▷ Update the learning parameters
10: **end for**

---

**Theoretical Analysis.** On the above optimization with infinite $\nu^*$, the Inverse Gamma distributions are converged to delta probability distribution, such that $\lim_{\nu^* \to \infty} q(\sigma^2) = \delta(\sigma^2)$, $\lim_{\nu^* \to \infty} p(\sigma^2) = \delta(\sigma^2)$. Hence, we can discover that the BNN from the ELBO generally loses its stochasticity after its training with the approximate posterior $q(w)$ via the zero convergence variance. For the hierarchical-ELBO, the tractable parameter $\psi$ is not required anymore during sampling the weight parameters in Fig. 1(b). This is because $\epsilon_{\sigma^2}$ is sampled to zero from delta distribution such that $\epsilon_{\sigma^2} \sim \delta(\epsilon_{\sigma^2}) = \lim_{\nu^* \to \infty} \mathcal{IG}(\nu, 1)$. Thus, only the Normal distribution $q(\mu) = \mathcal{N}(\mu_0, \sigma_0^2)$ is involved in sampling the weight parameters as follows: $w = \mu_0 + \sigma_0 \epsilon$. Here, according to the decomposition of the second term in Eq. (6), $q(\mu)$ is learned to follow the mean prior $p(\mu \mid \sigma^2)$ with the optimal parameter $\lambda^*$, such that $\lim_{\lambda \to \lambda^*} p(\mu \mid \sigma^2) = \mathcal{N}(0, k(\mu_0^2 + \sigma_0^2))$, where $k$ is statistically proven to be above one (see Appendix D). Then, the variance $\sigma_0^2$ of $q(\mu)$ follows the variance $k(\mu_0^2 + \sigma_0^2)$ of the mean prior, which is definitely larger than the variance $\sigma_0^2$. In short, training the BNN with the approximate conjugate prior leads to increase the change in the weight parameters, such that $w = \mu_0 + \sigma_0(\uparrow)\epsilon$. Therefore, we verify that the proposed method allows for the enhancement of the BNN's stochasticity through hierarchical-ELBO.

## 4 EXPERIMENTS

In this section, we validate adversarial robustness of the proposed method compared to strong baselines in various datasets. Since our method is inspired by adversarial training (Madry et al., 2018), BNN (Blundell et al., 2015), and adversarial-BNN (Liu et al., 2019), these three comparison methods are natural baseline methods.

**Implementation Details.** We conduct exhaustive experiments on not only a standard low dimensional dataset (CIFAR-10), but also more challenging datasets such as CIFAR-100, STL-10 (Coates et al., 2011), and Tiny-ImageNet (Le & Yang, 2015). STL-10 has 5,000 training images and 8,000 testing images with a size of 96×96 pixels. Tiny-ImageNet is a small subset of ImageNet dataset, containing 100,000 training images, 10,000 validation images, and 10,000 testing images separated in 200 different classes, whose dimensions are 64×64 pixels. Two vanilla networks, VGG and Model A[1], are adopted as our baseline networks. Model A is trained for STL-10 while the others are trained via VGG. Each baseline network is trained with $L_\infty$ 0.03 perturbation magnitude during adversarial training stage for comparing baseline methods and Ours. For validation, we apply the PGD attack and EOT-PGD attack to the test set, and adjust $L_\infty$ perturbation magnitude of the adversarial perturbation $\delta$ from 0 to 0.03 within the perturbation magnitude used in adversarial training.

**Computational complexity.** According to *Theoretical Analysis* above, the number of learning parameters in the proposed method is same number of those in the adversarial-BNN. Thus, there is no excessive computational complexity to implement the proposed method.

**Hyper-parameter.** To fairly validate adversarial robustness, we equally set hyper-parameter settings that learning rate equals to 0.01 with Adam (Kingma & Ba, 2015), and the learning parameters (mean and variance) are initialized to each $U(-1/\sqrt{c}, 1/\sqrt{c})$ ($c$ : channel number) and 0.1 (what

---

[1]Publicly available at `https://github.com/aaron-xichen/pytorch-playground/tree/master/stl10`

Table 1: The statistical summaries for the weight parameters compared with adversarial-BNN and Ours. We include mean, variance, and KL divergence (KLD). We describe the average of KLD for all layers in deep neural networks. Outperforming statistical results are marked in **bold**. The proposed method provides non-zero mean, higher variance (see Appendix F), and lower KLD.

| | CIFAR-10 | | | STL-10 | | | CIFAR-100 | | | Tiny-ImageNet | | |
|---|---|---|---|---|---|---|---|---|---|---|---|---|
| | Mean | Variance | KLD | Mean | Variance | KLD | Mean | Variance | KLD | Mean | Variance | KLD |
| adv-BNN | 0 | 0.073 | 6.25 | 0 | 0.085 | 1.28 | 0 | 0.073 | 8.87 | 0 | 0.069 | 15.53 |
| adv-Ours | 0.004 | **0.141** | **1.12** | 0.003 | **0.101** | **0.96** | 0.003 | **0.118** | **1.01** | 0.015 | **0.226** | **1.42** |

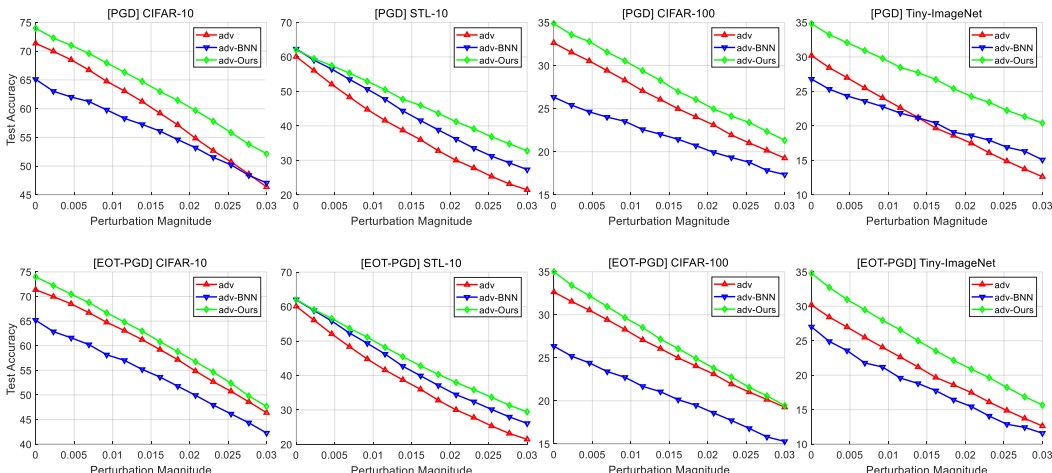

Figure 2: Comparison of classification accuracy under PGD attack (first row) and EOT-PGD attack (second row) to the $L_\infty$ perturbation magnitude on different datasets: CIFAR-10, STL-10, CIFAR-100, and Tiny-ImageNet.

we are interested in after training). For PGD attack, we set the number of steps $T$ to 10 in adversarial training stage and 30 in test stage, and set the step size $\eta$ to $\frac{2.3}{T}\delta$. For EOT-PGD attack, we equally use the hyper-parameters of PGD attack and set the number of samples to 10.

**Stochasticity and Regularization.** Tab. 1 shows comparison of statistical summaries for the weight parameters such as mean, variance and KL divergence in adversarial training with BNN and Ours. The summaries shows that naively putting BNN to adversarial training reduces the variance, and even fails to maintain the initial variance 0.1. In addition, the decreased variance causes higher KL divergence. This phenomenon shows an empirical evidence on the recall of the vanished stochasticity from adversarial training with BNN, where the broken BNN regularizer increases the KL divergence. Conversely, the proposed method shows comparatively higher variance (more increased variance than the initial variance) and lower KL divergence (6.7 times lower on average across all datasets). Therefore, we expect the proposed method to have a strong regularization effect so that the robustness discrepancy between adversarial-BNN and Ours becomes greater in large datasets with more classes such as CIFAR-100 and Tiny-ImageNet.

**Robustness under white-box PGD and EOT-PGD attack.** First, we compare adversarially trained baseline networks with baseline methods and Ours by adversarial robustness under the white-box PGD attack (first row in the Fig. 2) and EOT-PGD (second row in the Fig. 2) to the $L_\infty$ perturbation. As illustrated in this figure, adversarial-BNN has greater performance on STL-10 than naive adversarial training, but it has lower performances on the other datasets. While, the proposed method outperforms baseline methods to all datasets.

**Ablation studies.** We conduct three ablation studies: (1) attacking with higher perturbation magnitude ($\delta > 0.03$) than the perturbation magnitude of adversarial training, (2) adversarial training with large perturbation magnitude ($\delta = 0.06$) on WideResNet (Zagoruyko & Komodakis, 2016), and (3) comparison of clean training and adversarial training with BNN and Ours. The two ablation studies (1) and (2) are represented in Fig. 2 where the proposed method generally surpasses baseline methods, although it converges to the performances of baseline methods in much higher perturbation magnitude. For the ablation study (3), the statistical summaries of clean training on

Table 2: The statistical summaries between clean training without adversarial examples and adversarial training on WideResNet. The descriptions in this table are the same as Tab. 1.

| | Clean training ($\delta = 0$) | | | | | | | Adversarial training ($\delta = 0.06$) | | | | | |
| | CIFAR-10 | | | CIFAR-100 | | | | CIFAR-10 | | | CIFAR-100 | | |
| | Mean | Variance | KLD | Mean | Variance | KLD | | Mean | Variance | KLD | Mean | Variance | KLD |
|---|---|---|---|---|---|---|---|---|---|---|---|---|---|
| BNN | 0 | 0.077 | 5.82 | 0 | 0.065 | 23.59 | adv-BNN | 0 | 0.077 | 5.82 | 0 | 0.075 | 7.48 |
| Ours | 0.044 | **0.178** | **1.87** | 0.052 | **0.122** | **2.24** | adv-Ours | 0.015 | **0.192** | **1.01** | 0.031 | **0.146** | **1.42** |

(a) higher perturbation magnitude on Baseline networks ($\delta = 0.03$)    (b) adversarial training on WideResNet ($\delta = 0.06$)

Figure 3: Comparison of classification accuracy for two ablation studies: (a) attacking with higher perturbation magnitude ($\delta > 0.03$) than the maximum magnitude described in Fig. 2 on baseline networks (VGG, Model A) for CIFAR-10 and STL-10, (b) adversarial training with large perturbation magnitude ($\delta = 0.06$) on WideResNet for CIFAR-10 and CIFAR-100. These graphs are under PGD attack (first row) and EOT-PGD attack (second row) to the $L_\infty$ perturbation magnitude.

baseline networks (VGG, Model A) are illustrated in Appendix E. Further, Tab. 2 compares those of clean training and adversarial training with BNN and Ours on WideResNet. They show that both variances of BNN are lower than those of Ours. That is, this vanished stochasticity also happens in normal BNN when clean training. Here, we can analyze that the superior robustness of Ours does not come from the improvement of classification accuracy by itself.

**Discussion.** We tackle *why the proposed method is effective to EOT-like attack*. In general, a randomized network provides a stochastic tolerance obfuscating the gradient-based attack, which uses a single sample of the randomness. Especially, the BNN's stochasticity makes an attacker confused to incorrectly estimate the true gradient. But, EOT-like attack easily bypasses this stochasticity by taking the expectation over multiple weight parameters. However, our re-birth of BNN's stochasticity against EOT-like attack can be mathematically represented by Taylor series (see Appendix G). Now that it is possible to demonstrate the effectiveness of the proposed method against EOT-like attack, we can insist that the proposed method is not just obfuscating the gradient, but helping the deep neural networks to have strong adversarial robustness.

## 5 CONCLUSION

As mentioned in *Theoretical Analysis* at the Section 3, we have found that the BNN generally loses its stochasticity after its training with the BNN's posterior via the zero convergence variance. To solve it, we newly design the enhanced Bayesian regularizer with improved stochasticity through the hierarchical-ELBO under a concept of the conjugate prior. In addition, we have analyzed the improved stochasticity is effective to EOT-like attack. We believe the hierarchical-ELBO better approximates the true posterior against adversarial perturbations by the tight lower bound more than the ELBO, thus boosting the adversarial robustness. Further progress can be made by investigating the hierarchical-ELBO to explore generative models, such as Variational Auto-Encoder (VAE) and Generative Adversarial Network (GAN) for scalable machine learning tasks.

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

## A  DECOMPOSITION OF KL DIVERGENCE IN HIERARCHICAL-ELBO

The approximate conjugate prior $q(\mu, \sigma^2)$ can be independently factorized into the Normal distribution $q(\mu)$ and the Inverse Gamma distribution $q(\sigma^2)$ by the mean-field approximation. The hierarchical prior $p(\mu, \sigma^2)$ is factorized into the mean prior $p(\mu \mid \sigma^2)$ and the variance prior $p(\sigma^2)$. The KL divergence in the hierarchical-ELBO can be decomposed as follows:

$$\mathcal{D}_{KL}(q(\mu, \sigma^2) \mid\mid p(\mu, \sigma^2))$$

$$= \iint q(\mu, \sigma^2) \log \frac{q(\mu, \sigma^2)}{p(\mu, \sigma^2)} d\mu d\sigma^2$$

$$= \iint q(\mu) q(\sigma^2) \log \frac{q(\mu) q(\sigma^2)}{p(\mu^2 \mid \sigma^2) p(\sigma^2)} d\mu d\sigma^2$$

$$= \iint q(\mu) q(\sigma^2) \log \frac{q(\mu)}{p(\mu \mid \sigma^2)} d\mu d\sigma^2 + \iint q(\mu) q(\sigma^2) \log \frac{q(\sigma^2)}{p(\sigma^2)} d\mu d\sigma^2$$

$$= \int q(\sigma^2) \left[ \int q(\mu) \log \frac{q(\mu)}{p(\mu \mid \sigma^2)} d\mu \right] d\sigma^2 + \int q(\sigma^2) \log \frac{q(\sigma^2)}{p(\sigma^2)} d\sigma^2$$

$$= E_{\sigma^2 \sim q(\sigma^2)} \left[ \mathcal{D}_{KL}(q(\mu) \mid\mid p(\mu \mid \sigma^2)) \right] + \mathcal{D}_{KL}(q(\sigma^2) \mid\mid p(\sigma^2)).$$

(9)

## B  FULL EXPANSION OF KL DIVERGENCE IN HIERARCHICAL-ELBO

The first term in the KL divergence at the hierarchical-ELBO can be written as:

$$E_{\sigma^2 \sim q(\sigma^2)} \left[ \mathcal{D}_{KL}(q(\mu) \mid\mid p(\mu \mid \sigma^2)) \right]$$

$$= \sum_{j=1}^{M} E_{\sigma_j^2 \sim q(\sigma_j^2)} \left[ \mathcal{D}_{KL}(q(\mu_j) \mid\mid p(\mu_j \mid \sigma_j^2)) \right] \quad \textit{(by Lemma B.4)}$$

$$= \sum_{j=1}^{M} E_{\sigma_j^2 \sim q(\sigma_j^2)} \left[ \frac{\sigma_{0,j}^2 + \mu_{0,j}^2}{2\lambda_j^{-1}\sigma_j^2} + \frac{1}{2} \log \frac{\lambda_j^{-1}\sigma_j^2}{\sigma_{0,j}^2} - \frac{1}{2} \right] \quad \textit{(by Lemma B.1)}$$

$$= \sum_{j=1}^{M} \left[ \nu_j \frac{\sigma_{0,j}^2 + \mu_{0,j}^2}{2\lambda_j^{-1}\psi_j} + \frac{1}{2} \log \frac{\lambda_j^{-1}\psi_j}{\sigma_{0,j}^2} - \frac{1}{2} \frac{\Gamma'(\nu_j)}{\Gamma(\nu_j)} - \frac{1}{2} \right], \quad \textit{(by Lemma B.2)}$$

(10)

where

$$q(\mu_j) = \mathcal{N}(\mu_j \mid \mu_{0,j}, \sigma_{0,j}^2), \quad p(\mu_j \mid \sigma_j^2) = \mathcal{N}(\mu_j \mid 0, \lambda_i^{-1}\sigma_j^2).$$

(11)

The second term in the KL divergence at the hierarchical-ELBO can be written as:

$$\mathcal{D}_{KL}(q(\sigma^2) \;||\; p(\sigma^2))$$

$$= \sum_{j=1}^{M} \mathcal{D}_{KL}(q(\sigma_j^2) \;||\; p(\sigma_j^2)) \quad \text{(by Lemma B.4)} \tag{12}$$

$$= \sum_{j=1}^{M} \left[ \nu_j \log \psi_j - \nu_j \frac{\psi_j - 1}{\psi_j} \right] \quad \text{(by Lemma B.3)}$$

where

$$q(\sigma_j^2) = \mathcal{IG}(\sigma_j^2 \mid \nu_j, \psi_j), \quad p(\sigma_j^2) = \mathcal{IG}(\sigma_j^2 \mid \nu_j, 1). \tag{13}$$

Full expansion of the KL divergence can be computed by the summation of the above two terms Eq. (10) and Eq. (12). The formulation can be written as:

$$\mathcal{D}_{KL}(q(\mu, \sigma^2) \;||\; p(\mu, \sigma^2)) = \frac{1}{M} \sum_{j=1}^{M} [\nu_j \frac{\lambda_j(\sigma_{0,j}^2 + \mu_{0,j}^2) + 2}{2\psi_j} + \log \frac{\psi_j^{\frac{2\nu_j+1}{2}}}{(\lambda_i \sigma_{0,j}^2)^{\frac{1}{2}}} - \frac{2\nu_j + 1}{2} - \frac{1}{2} \frac{\Gamma'(\nu_j)}{\Gamma(\nu_j)}], \tag{14}$$

**Lemma B.1.** *KL divergence between two Gaussian distributions.*

$$\mathcal{D}_{KL}(q(x)||p(x)) = \frac{\sigma_q^2 + (\mu_q - \mu_p)^2}{2\sigma_p^2} + \frac{1}{2} \log \frac{\sigma_p^2}{\sigma_q^2} - \frac{1}{2},$$

*where*

$$q(x) = \mathcal{N}(x \mid \mu_q, \sigma_q^2), \quad p(x) = \mathcal{N}(x \mid \mu_p, \sigma_p^2).$$

**Lemma B.2.** *The expected Inverse Gamma distribution*

$$E_{x \sim q(x)} \left[ \frac{1}{x} \right] = \frac{\nu}{\psi}, \quad E_{x \sim q(x)} [\log x] = \log \psi - \frac{\Gamma'(\nu)}{\Gamma(\nu)},$$

*where*

$$q(x) = \mathcal{IG}(x \mid \nu, \psi) = \frac{\psi^\nu}{\Gamma(\nu)} x^{-(\nu+1)} \exp\left(-\psi x^{-1}\right)$$

**Lemma B.3.** *KL divergence between two Inverse Gamma distributions*

$$\mathcal{D}_{KL}(q(x)||p(x)) = \int q(x) \log \frac{\frac{\psi_q^\nu}{\Gamma(\nu)} x^{-(\nu+1)} \exp\left(-\psi_q x^{-1}\right)}{\frac{\psi_p^\nu}{\Gamma(\nu)} x^{-(\nu+1)} \exp\left(-\psi_p x^{-1}\right)} dx$$

$$= \int q(x) \log \left[\frac{\psi_q^\nu}{\psi_p^\nu} \exp\left(-(\psi_q - \psi_p)x^{-1}\right)\right] dx$$

$$= \int q(x) \left[\nu \log \frac{\psi_q}{\psi_p} - (\psi_q - \psi_p)x^{-1}\right] dx$$

$$= \nu \log \frac{\psi_q}{\psi_p} - (\psi_q - \psi_p) \int \frac{q(x)}{x} dx$$

$$= \nu \log \frac{\psi_q}{\psi_p} - \frac{\nu(\psi_q - \psi_p)}{\psi_q}, \quad \textit{(by Lemma B.2)}$$

*where*

$$q(x) = \mathcal{IG}(x \mid \nu, \psi_q), \quad p(x) = \mathcal{IG}(x \mid \nu, \psi_p).$$

**Lemma B.4.** *KL divergence with independent random variables*

$$\mathcal{D}_{KL}(q(x)||p(x)) = \iint \cdots \int q(x) \log \frac{q(x)}{p(x)} dx_1 dx_2 \cdots dx_M$$

$$= \iint \cdots \int q(x) \log \left[\prod_{j=1}^{M} \frac{q(x_j)}{p(x_j)}\right] dx_1 dx_2 \cdots dx_M$$

$$= \sum_{j=1}^{M} \iint \cdots \int q(x) \log \frac{q(x_j)}{p(x_j)} dx_1 dx_2 \cdots dx_M$$

$$= \sum_{j=1}^{M} \underbrace{\int q(x_j) \log \frac{q(x_j)}{p(x_j)} dx_j}_{\mathcal{D}_{KL}(q(x_j)||p(x_j))} \times \underbrace{\prod_{\substack{i=1 \\ i \neq j}}^{M} \int q(x_i) dx_i}_{1}$$

$$= \sum_{j=1}^{M} \mathcal{D}_{KL}(q(x_j)||p(x_j)),$$

*where*

$$x = (x_1, x_2, \cdots, x_M), \quad q(x) = \prod_{j=1}^{M} q(x_i), \quad p(x) = \prod_{j=1}^{M} p(x_i).$$

## C   FUNCTION ANALYSIS

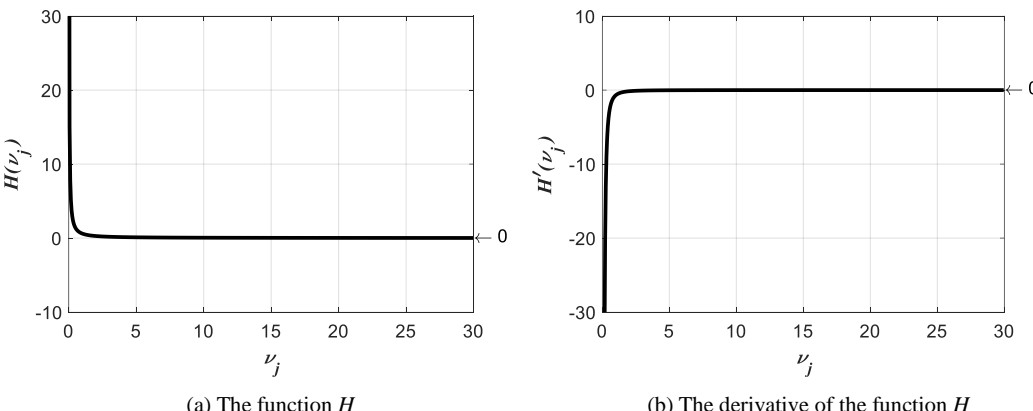

(a) The function $H$

(b) The derivative of the function $H$

Figure 4: Function analysis for (a) the function $H$, and (b) the derivative of the function $H$,

where

$$H(\nu_j) = \frac{\nu_j}{2}\frac{d}{d\nu_j}G(\nu_j) - \frac{1}{2}G(\nu_j) + \frac{1}{2}\log\nu_j - \frac{1}{2}$$

$$= \frac{1}{2}\sum_{k=0}^{\infty}\frac{\nu_j}{(\nu_j+k)^2} - \frac{\Gamma'(\nu_j)}{\Gamma(\nu_j)} + \frac{1}{2}\log\nu_j - \frac{1}{2} > 0$$

(15)

and

$$\lim_{\nu_j \to \infty} H(\nu_j) = 0^+.$$

(16)

where

$$H'(\nu_j) = \frac{\nu_j}{2}\frac{d^2}{d\nu_j^2}G(\nu_i) + \frac{1}{2\nu_j}$$

$$= -\sum_{k=0}^{\infty}\frac{\nu_j}{(\nu_j+k)^3} + \frac{1}{2\nu_j} < 0$$

(17)

and

$$\lim_{\nu_j \to \infty} H'(\nu_j) = 0^-.$$

(18)

## D   PROOF OF $k$ VALUE

The Normal distribution $q(\mu) = \mathcal{N}(\mu_0, \sigma_0^2)$ is learned to follow the mean prior $p(\mu \mid \sigma^2) = \mathcal{N}(0, (\lambda^*)^{-1}\sigma^2)$, with the optimal value $\lambda^* = \frac{\psi}{\nu(\mu_0^2+\sigma_0^2)}$. The mean prior's variance equals to $\frac{\nu\sigma^2}{\psi}(\mu_0^2 + \sigma_0^2)$. Here, $\epsilon_{\sigma^2} = \frac{\sigma^2}{\psi}$ is sampled from $\mathcal{IG}(\nu, 1)$, as depicted in the Fig. 1(b). Thus, the variance is written as: $\nu\epsilon_{\sigma^2}(\mu_0^2 + \sigma_0^2)$. Now, we can statistically estimate $\nu\epsilon_{\sigma^2}$ by the expectation over $\epsilon_{\sigma^2} \sim \mathcal{IG}(\nu, 1)$. Then, it satisfies $\mathbb{E}_{\epsilon_{\sigma^2}}[\nu\epsilon_{\sigma^2}] = \frac{\nu}{\nu-1} > 1$. Hence, the mean prior $p(\mu \mid \sigma^2)$ can be designed to $\mathcal{N}(0, k(\mu_0^2 + \sigma_0^2))$, where $k > 1$.

# E CLEAN TRAINING WITHOUT ADVERSARIAL EXAMPLES

Table 3: On the clean training, the statistical summaries for the weight parameters compared with BNN and Ours. We include mean, variance, and KL divergence (KLD). We describe the average of KLD for all layers in deep neural networks. Outperforming statistical results are marked in **bold**.

|  | CIFAR-10 | | | STL-10 | | | CIFAR-100 | | | Tiny-ImageNet | | |
|---|---|---|---|---|---|---|---|---|---|---|---|---|
|  | Mean | Variance | KLD | Mean | Variance | KLD | Mean | Variance | KLD | Mean | Variance | KLD |
| BNN | 0 | 0.071 | 7.7 | 0 | 0.051 | 2.5 | 0 | 0.068 | 15.4 | 0 | 0.064 | 26.7 |
| Ours | 0.003 | **0.129** | **1.0** | 0.002 | **0.092** | **1.0** | 0.007 | **0.109** | **1.4** | 0.028 | **0.211** | **1.8** |

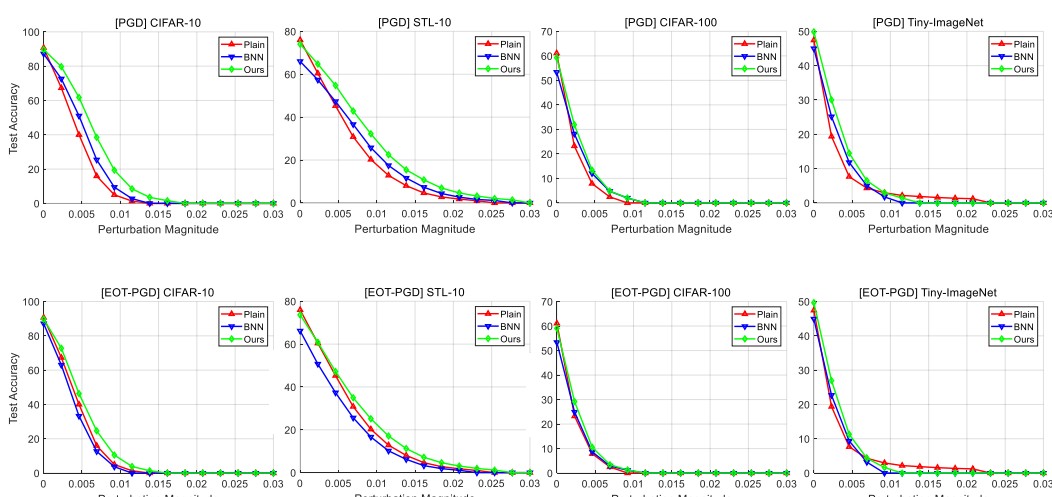

Figure 5: Comparison of classification accuracy without adversarial training. These graphs are under PGD attack (first row) and EOT-PGD attack (second row) to the $L_\infty$ perturbation magnitude on different datasets: CIFAR-10, STL-10, CIFAR-100, and Tiny-ImageNet.

# F NON-ZERO MEAN AND HIGHER VARIANCE

**Non-zero Mean.** According to the experiment results in Tab. 1-3, the mean learned from adversarial training with BNN follows zero, while the mean from the proposed method follows non-zero. Analyzing KL divergences between ELBO and hierarchical-ELBO can provide a clue to this phenomenon. The relationship of their KL divergences can be written as follows:

$$\underbrace{\mathcal{D}_{KL}(q(\mu, \sigma^2) \,||\, p(\mu, \sigma^2))}_{\text{for hierarchical-ELBO}} < \underbrace{\mathcal{D}_{KL}(q(w) \,||\, p(w))}_{\text{for ELBO}}. \tag{19}$$

The left KL divergence above represents the proposed method to better approximate the true posterior by the tight lower bound of hierarchical-ELBO, and the right one denotes the BNN regularizer in ELBO during adversarial training with BNN. Through the optimization described in the Section 3, the proposed method can be much smaller than the BNN regularizer. Based on this relationship of Eq. (19), it is not required that hierarchical-ELBO concentrates on learning its KL divergence due to already optimized strong regularization. In contrast, ELBO more tries to reduce its KL divergence to improve regularization effect through the zero mean, the purpose of which is addressing weak regularization effect attributed by the vanished stochasticity. At this time, the proposed method relatively more focuses to learn the expected log-likelihood in the hierarchical-ELBO, where the mean does not have to be zero. Instead, the mean is adaptively learned according to the characteristics of the datasets.

**Higher Variance.**    As also shown in Tab. 1-3, the proposed method promotes higher variance compared to the initial variance 0.1. While, adversarial training with BNN has lower variance causing the lack of its stochasticity. We contemplate whether the stochasticity and higher variance purely affects the adversarial robustness of the deep neural networks. To examine this impact, we handle the magnitude of the derivative to the log-likelihood over the weight parameters, which can be written as follows:

$$\left\| \frac{d}{dw} \log p(y \mid x^{adv}, w) \right\| = \left\| \frac{1}{\sigma} \frac{d}{d\epsilon} \log p(y \mid x^{adv}, \mu + \sigma\epsilon) \right\| \tag{20}$$

where $w = \mu + \sigma\epsilon$, such that $\epsilon \sim \mathcal{N}(0, 1)$. Note that for mathematical convenience, the denoted $\mu$ and $\sigma^2$ correspond to the mean and the variance respectively in both adversarial training with BNN and Ours.

Eq. (20) explicates the magnitude of the change in the cross-entropy as the weight parameters change from the variation of $\epsilon$, which can be represented to: $dw = \sigma d\epsilon$. Once the variance gets much smaller due to the vanished stochasticity, the magnitude becomes larger. That is, prediction results of the deep neural networks differ significantly from the slight change of the weight parameters, such that $w = \mu + \sigma(\approx 0)\epsilon$. The large change in cross-entropy makes ELBO inconsistent, and it loses adversarial robustness. In contrast, once the variance from the proposed method gets much larger, the prediction results are robust to the large change of the weight parameters, where the magnitude becomes smaller. Accordingly, Eq. (20) guides the proposed method to keep the maximum hierarchical-ELBO consistent by the higher variance, thus better obtaining well-posed inference to boost adversarial robustness. Consequently, we verify that the higher variance has a positive impact on adversarial robustness by itself with large BNN's stochasticity.

## G    ROBUSTNESS OF PROPOSED METHOD AGAINST EOT-LIKE ATTACK

Let $T(w) = \nabla_x J(f_w(x), y)$ where $w$ denotes the weight parameters $w = \mu + \sigma\epsilon$, such that $\epsilon \sim \mathcal{N}(0, 1)$. Note that for mathematical convenience, the denoted $\mu$ and $\sigma^2$ correspond to the mean and the variance respectively in both adversarial training with BNN and Ours.

Expectation over Transformation (EOT) attack is well-known adaptive attack targeting randomized neural networks. The EOT-like attack easily bypasses the stochasticity of randomized classifiers by taking the expectation over the multiple weight parameters to compute the actual gradient. The formulation of the expected $T$ over the weight parameters can be written as follows:

$$\mathbb{E}_w[T(w)] = \mathbb{E}_\epsilon[T(\mu + \sigma\epsilon)]. \tag{21}$$

According to Taylor series, the above equation can be written as follows:

$$\mathbb{E}_\epsilon[T(\mu + \sigma\epsilon)] = \mathbb{E}_\epsilon[T(\mu) + \sigma\epsilon T'(\mu) + \frac{(\sigma\epsilon)^2}{2!}T''(\mu) + \cdots]$$
$$= \underbrace{\mathbb{E}_\epsilon[T(\mu)]}_{\text{actual gradient}} + \mathbb{E}_\epsilon[\sigma\epsilon T'(\mu)] + \mathbb{E}_\epsilon[\frac{(\sigma\epsilon)^2}{2!}T''(\mu)] + \cdots . \tag{22}$$

For naive adversarial training ($\sigma$ is not used), the expected $T$ can be written as:

$$\mathbb{E}_\epsilon[T(\mu)] = T(\mu). \tag{23}$$

For adversarial training with BNN, the variance $\sigma^2$ of the approximate posterior $q(w)$ becomes to be reduced, so Taylor series ends up with first-order approximation. Eq. (22) can be written as:

$$\mathbb{E}_\epsilon[T(\mu + \sigma\epsilon)] = \mathbb{E}_\epsilon[T(\mu)] + \mathbb{E}_\epsilon[\sigma\epsilon T'(\mu)] = T(\mu), \tag{24}$$

which equals to the result of the expected $T$ from naive adversarial training in Eq. (23). On the other hand, the proposed method can enhance the BNN's stochasticity with the large enough variance $\sigma^2$. Hence, the remaining higher order terms of Taylor series in Eq. (22) still survive at even degree, which can be written as:

$$\mathbb{E}_\epsilon[T(\mu + \sigma\epsilon)] = \mathbb{E}_\epsilon[T(\mu) + \sigma\epsilon T'(\mu) + \frac{(\sigma\epsilon)^2}{2!}T''(\mu) + \cdots]$$

$$= \underbrace{T(\mu)}_{\text{actual gradient}} + \underbrace{\frac{\sigma^2}{2}T''(\mu) + \cdots}_{\text{stochastic tolerant terms}} . \tag{25}$$

The remaining terms of Taylor series make a difference from the actual gradient as opposed to adversarial training and adversarial-BNN. Here, these terms retain stochastic tolerance against EOT-like attack, thus bringing in the improvement of adversarial robustness as the effectiveness of the proposed method.

