# OpenReview forum: "Towards Adversarial Robustness of Bayesian Neural Network through Hierarchical Variational Inference"
_ICLR.cc/2021/Conference — Reject_

### Official Review · AnonReviewer1 · 2020-10-28
**A sound method with unclear experimental settings**

**Rating:** 5
**Confidence:** 4

**Review:**

This paper studies the adversarial robustness of DNNs with Bayesian neural networks. Although BNN has been integrated with adversarial training for better robustness, this paper argues that the previous method lacks the stochasticity (i.e., the posterior tends to have zero variance), thus limiting the robustness performance. In this paper, a new hierarchical variational inference is proposed to enhance the robustness when integrating with adversarial training. The proposed method is presented well. The experiments show the effectiveness of the proposed method.

Besides, I have few concerns about this paper.

1. This paper argues that the previous method (ADV-BNN) learns the posterior distribution that has near zero variance, but does not analyze why ADV-BNN causes this phenomenon. Does normal BNN models (without adversarial training) also have this issue?

2. The experimental settings are unclear in the context. The parameters of PGD and EOT-PGD are not stated (e.g., number of steps, step size, number of samples in EOT, etc.). Therefore, it is hard to judge the significance of the results.

3. This paper lacks the comparison with the state-of-the-art methods. A common practice is to use Wide ResNet models for adversarial training and set the L_infty norm of perturbation as 16/255. I suggest the authors to compare with the public adversarial training models.

---

> ### Author Response · Authors · 2020-11-20
> **Responses for AnonReviewer1**
>
> We are grateful for your comments and suggestions as they helped us further clarify the reason why the near zero variance occurs in the adv-BNN and normal BNN. We also appreciate for your valuable comments on the experimental settings and proper comparison with state-of-the-art models.
> ***
> **Q1. This paper argues that the previous method (adv-BNN) learns the posterior distribution that has near zero variance, but does not analyze why adv-BNN causes this phenomenon. Does normal BNN models (without adversarial training) also have this issue?**
>
> A1. As you pointed out, we modified and concretized the mathematical analysis on the near zero variance phenomenon at Theoretical Analysis section in page 7. Furthermore, for your question regarding normal BNN models, you can refer to the Appendix E (clean training without adversarial examples), the statistical summaries in Clean training at Table 2, and Table 3 empirically showed normal BNN also has same phenomenon of the vanished stochasticity.
> ***
> **Q2. The experimental settings are unclear in the context. The parameters of PGD and EOT-PGD are not stated (e.g., number of steps, step size, number of samples in EOT, etc.). Therefore, it is hard to judge the significance of the results.**
>
> A2. We newly changed and added detailed information of the parameters at Hyper-parameter section in page 7. For PGD attack, we set the number of steps T to 10 in adversarial training stage and 30 in test stage, and set the step size $\eta$ to $\frac{2.3}{T}\delta$. For EOT-PGD attack, we equally use the hyper-parameters of PGD attack and set the number of samples to 10.
> ***
> **Q3. This paper lacks the comparison with the state-of-the-art methods. A common practice is to use Wide ResNet models for adversarial training and set the $L_\infty$ norm of perturbation as 16/255. I suggest the authors to compare with the public adversarial training models.**
>
> A3. We added ablation study for comparison with Wide ResNet models in page 8-9. You can check the modification on the newly uploaded paper. As you already mentioned, since Madry et al. (2017) [1] used Wide ResNet [2] which is a residual convolutional neural network consisting of a number of residual units and a fully connected layer [1], we performed the additional experiment for CIFAR10 and CIFAR100. We trained and compared three methods on the Wide ResNet models with adversarial training and set the $L_\infty$ norm of perturbation as 16/255 as you suggested. We summarized results of the additional experiments on Wide ResNet against PGD attack as seen in the following Tables. The results demonstrate that the proposed method outperforms the baseline methods even on the state-of-the-art model as well. The detailed performance against EOT attack and statistical summaries of adv-BNN and adv-Ours on Wide ResNet were added in Figure 3 and Table 2 of the newly uploaded paper.
>
> * PGD attack on WideResNet trained with $\delta = 16/255$ for CIFAR10
> | $\delta$	|   0   	| 0.018 	| 0.031 	| 0.043 	| 0.062 	|  0.08 	|
> |:--------:	|:-----:	|:-----:	|:-----:	|:-----:	|:-----:	|:-----:	|
> |    adv   	| 57.56 	| 49.52 	| 44.01 	| 38.17 	| 29.12 	| 21.14 	|
> |  adv-BNN 	| 53.30 	| 46.49 	| 42.36 	| 38.39 	| 32.38 	| 26.47 	|
> | adv-Ours 	| 66.12 	|  58.4 	|  52.7 	| 46.55 	| 37.38 	| 28.14 	|
>
> * PGD attack on WideResNet trained with $\delta = 16/255$ for CIFAR100
> | $\delta$	|   0   	| 0.018 	| 0.031 	| 0.043 	| 0.062 	|  0.08 	|
> |:--------:	|:-----:	|:-----:	|:-----:	|:-----:	|:-----:	|:-----:	|
> |    adv   	| 33.40 	| 27.03 	| 23.21 	| 19.82 	| 14.55 	| 10.20 	|
> |  adv-BNN 	| 26.10 	| 21.45 	| 19.12 	| 16.69 	| 13.57 	| 10.51 	|
> | adv-Ours 	| 37.00 	| 29.84 	| 25.85 	| 21.79 	| 16.44 	| 12.05 	|
>
> ***
> References
>
>  * [1] Madry, A., Makelov, A., Schmidt, L., Tsipras, D., & Vladu, A. (2017). Towards deep learning models resistant to adversarial attacks. arXiv preprint arXiv:1706.06083.
>
>  * [2] Zagoruyko, S., & Komodakis, N. (2016). Wide residual networks. arXiv preprint arXiv:1605.07146.

---

### Official Review · AnonReviewer2 · 2020-10-28
**An extension to Adv-BNN to improve the robustness of BNN**

**Rating:** 6
**Confidence:** 3

**Review:**

Summary:

This paper presents a new adversarial training for BNNs with variational inference (VI). Specifically, Adv-BNN training of Liu et al. 2019 uses a standard normal prior for VI of BNNs.  The paper observes that the above method may have vanished stochasticity that reduces the robustness and the proposed method extends it with a conjugate prior constructed by a normal distribution and an inverse gamma distribution. This extension results in a stronger regularisation of the weights of BNNs, which can enhance the robustness against adv attacks and leads to a hierarchical inference. The method is reported to have better performance than vanilla adv training and adv-BNN training on several benchmark datasets.

Pros:
- It is an interesting and motivative observation that AdV-BNN has vanished stochasticity issue, which is important for BNNs.

- The proposed method is a straightforward way to address the vanished stochasticity issue.

- The results in Table 1 are intuitive, which directly shows the proposed method has more stochasticity than Adv-BNNs.

Cons:

- It is concerning that the reported performance of Adv-BNN in this paper has a significant difference than that reported in the original paper. In this paper, Adv-BNN performs worse than the vanilla adv training with a large margin, which is a bit surprised. Therefore, it is unclear whether it is Adv-BNN not working or it is the settings/implementations of this paper having something wrong. Given this fact, the performance reported in this paper seems to be ungrounded. It is hard to justify the true performance advantage reported in this paper.

- The approach might be less related to the topic of "hierarchical inference", as it's only replacing the standard Gaussian with a normal-inverse-gamma distribution, which only affects the KL divergence in this case, where there are no intermediate variables are inferred.

- Minor: Some of the notations and equations are a bit unclear. For example, q has been used to denote the posterior but it denotes the prior in Eq. (5)

--------------------------------------------------------------------------------------------------------------------------------------------------

The author response addresses my major concern on the experimental results. Therefore, I have updated my rating from 5 to 6.

---

> ### Author Response · Authors · 2020-11-20
> **Responses for AnonReviewer2 (cont')**
>
> We thank you for your insightful and helpful comments. As AnonReviewer3 also noted, your major concern is the difference between reported performance of adv-BNN in the original paper and ours. Also, we appreciate your comments on the topic of hierarchical inference for clarifying the advantages of the proposed method.
>
> ***
>
> **Q1. It is concerning that the reported performance of adv-BNN in this paper has a significant difference than that reported in the original paper. In this paper, adv-BNN performs worse than the vanilla adv training with a large margin, which is a bit surprised. Therefore, it is unclear whether it is adv-BNN not working or it is the settings/implementations of this paper having something wrong. Given this fact, the performance reported in this paper seems to be ungrounded. It is hard to justify the true performance advantage reported in this paper.**
>
> A1. In a nutshell, we used different experimental settings in the scope of layers with Bayesian inference applied. Zimmermann (2019) [1] pointed out the open source code of the original adv-BNN did not use the full Bayesian inference to the whole layer ([https://github.com/xuanqing94/BayesianDefense/issues/5](https://github.com/xuanqing94/BayesianDefense/issues/5)). He noted that although the authors of adv-BNN calculated the KL divergence for all layers in the network in their paper, but they did not use the sum of the KL divergences for all layers. Rather, the original code actually used just the KL divergence of the last layer ([https://github.com/xuanqing94/BayesianDefense/blob/master/models/vgg_vi.py#L39](https://github.com/xuanqing94/BayesianDefense/blob/master/models/vgg_vi.py#L39)). Thus, according to the public open source code of adv-BNN, their proposed regularization is only applied to the last layer during training and not the entire network. Since our proposed regularization is theoretically proved and described in the sense of applying Bayesian inference to the whole layer in the submitted paper, we utilized hierarchical Bayesian inference in the entire network, not just the last layer.
>
> To avoid controversy, we can start from currently available open source code for the adv-BNN ([https://github.com/xuanqing94/BayesianDefense](https://github.com/xuanqing94/BayesianDefense)). When we didn’t change anything in the open source code of the original adv-BNN, we were able to reproduce a performance similar to the original adv-BNN as below. However, switching to using the sum of KL divergence for all layers did not allow us to achieve the performance in the original paper.
>
> When we used the calculation of only the last layer KL divergence for the proposed method, it still outperformed adv-BNN in small margins with the calculation for the last layer KL divergence. We think this small difference comes from the different Bayesian inference applying only the last layer. If we use the full Bayesian inference to the whole layer, then the difference becomes huge.
>
> In short, the described Figure 2-4 (also even Figure 3 newly added in updated paper for ablation study) were fairly compared figures with adv, adv-BNN, and adv-Ours on the equal evaluation settings where we utilized the whole layer summation of KL divergence for full Bayesian.
>
> * Comparison between adv-BNN and adv-Ours on CIFAR10
> | $\delta$ 	| adv-BNN (origianl paper) 	| adv-BNN (KLD Last Only) 	| adv-BNN (KLD Sum) 	| adv-Ours (KLD Last Only) 	| adv-Ours (KLD Sum) 	|
> |:--------:	|:------------------------:	|:-----------------------:	|:-----------------:	|:------------------------:	|:------------------:	|
> |     0    	|           79.7           	|          79.94          	|        65.10       	|           81.91          	|        74.00       	|
> |   0.015  	|           68.7           	|          68.52          	|       56.00       	|           69.67          	|        64.73       	|
> |   0.035  	|           45.4           	|          46.13          	|       47.08       	|           48.18          	|        52.16       	|
> |   0.055  	|           26.9           	|          27.35          	|       30.23       	|           27.71          	|        32.03       	|
> |   0.07            |          18.6            |          18.23                   |      20.67              |           18.64                  |        22.64           |
>
> ***
>
> Reference
> * [1] Zimmermann, R. S. (2019). Comment on" Adv-BNN: Improved Adversarial Defense through Robust Bayesian Neural Network". arXiv preprint arXiv:1907.00895.

---

> > ### Author Response · Authors · 2020-11-20
> > **Responses for AnonReviewer2 (cont')**
> >
> > **Q2. The approach might be less related to the topic of "hierarchical inference", as it's only replacing the standard Gaussian with a normal-inverse-gamma distribution, which only affects the KL divergence in this case, where there are no intermediate variables are inferred.**
> >
> > A2. Figure 1(b) represents hierarchical inference to sample weight parameters with $\mu_0$, $\sigma_0$, $\psi$, and $\nu$ in the approximate conjugate prior, such that $w=\mu(=\mu_{0}+\sigma_{0}\epsilon_{\mu})+\sigma (=(\psi\epsilon_{\sigma^2})^{1/2})\epsilon$. By doing the optimization described in Section 3, Theoretical Analysis in page 7 demonstrated that the variance converges to zero: $\sigma^2 \approx 0$, then it satisfies $w=\mu$. Here, the mean $\mu$ is sampled from the Normal distribution $q(\mu)$ in the approximate conjugate prior $q(\mu, \sigma^2)$ such that $w=\mu=\mu_0+\sigma_0\epsilon_{\mu}$. Although the formulations of sampling the weight parameters are explicitly same between variational inference and hierarchical variational inference, $\mu$ and $\mu_0$ are implicitly different parameters because $\mu$ denotes the mean in the approximate posterior for variational inference, and $\mu_0$ indicates the mean prior knowledge in the approximate conjugate prior for hierarchical variational inference. In short, the proposed method deals with hierarchical parameters to sample the weight parameters where $\mu$ and $\mu_0$ are different in nature. Furthermore, this zero variance in the conjugate view has an advantage of no additional computational complexity compared with adversarial-BNN, while achieving more robust performance.
> > ***
> > **Q3. Minor: Some of the notations and equations are a bit unclear. For example, q has been used to denote the posterior but it denotes the prior in Eq. (5).**
> >
> > A3. We modified the mathematical terms to clearly deliver them. As the approximate posterior $q(w)$ approximates the true posterior, we modified previous expression of $q(\mu, \sigma^2)$ to denote it as an approximate conjugate prior which is approximating the conjugate prior as a same distribution family of the true posterior. And, our original intention of denoted $q$ more focuses the meaning of "approximate" rather than whether it is posterior or prior. Updated version has been uploaded. And, you can check our modification that you were concerned with before.

---

### Official Review · AnonReviewer4 · 2020-10-28
**Nice results and important direction, but too many misnomers. [Edited Post-Rebuttal]**

**Rating:** 5
**Confidence:** 4

**Review:**

In this paper the authors study the adversarial robustness of BNNs on large scale datasets. BNNs have been shown to be a more robust learning paradigm due to their uncertainty/stochasticity. Given the empirical observation that adversarially trained BNN posterior variances converge to zero (which the authors need to do much more to show as this is not a well-established phenomena), the authors propose a hierarchical prior where they put a prior over the parameters of the Gaussian prior normally used  in mean-field variational inference. The authors show that performing approximate inference with a hierarchical prior leads to an increased variational posterior variance, which the authors claim is correlated to the observation of increased adversarial robustness.

I think the empirical results are solid, and the direction of the paper is ultimately an important one. I truly encourage the authors to continue to pursue this topic.

Unfortunately, this paper is severely handicapped by its lack of clarity in terms of accurate contextualization, and its pervasive use of misnomers.  These misnomers are so prevalent that they would certainly lead uninformed readers to incorrect conclusions about Bayesian deep learning.

* On the “true posterior:”  There are many places throughout the paper where the authors discuss/reference the computation of the true posterior for a Bayesian deep neural network. This can only be done by exact Bayesian inference. That includes (1) proper marginalization over the space of parameters (2) normalization wrt p(X). Given a DNN with non-linear activations, computation of the true posterior is intractable. Despite this, the authors claim that one can infer the true posterior via variation inference methods (bottom of pg 1). Variational inference makes a closed form approximation of the posterior that one tries to learn the parameters of. Even learning the optimal parameters does not guarantee convergence, outside of the case of conjugation.  Further on this, computing the true adversarial posterior is even more intractable given that the intractability of computing the optimal adversarial example compounds the issue of performing exact Bayesian inference.

* Conjugate priors and hierarchical priors are distinct under the Bayesian framework. Despite this, the authors name their hierarchical prior the “conjugate” prior. In this work, the authors suggest placing a prior distribution over the parameters of their prior distribution (i.e. a hierarchical prior), yet call it a conjugate prior. A conjugate prior in the standard Bayesian literature is a prior which is known to be in the same family as the true posterior. It is not known, and likely not true, that for general approximate Bayesian neural networks (e.g. mean-field approximations) the true or approximate posterior is Gaussian. Thus, it is likely false to call a mean-field prior approximation a “conjugate” prior.

 * Robust Optimization is the special case of adversarial training where only adversarial data is used. The authors conflate adversarial optimization (optimization with respect to an adversarial objective) with robust optimization which has a rich history in optimization prior to its application to deep learning. The end of section 2.1 should have its terminology corrected.

* On the notion of ‘regularization’ in Bayesian deep learning. In several places the authors refer to the regularization term of the ELBO objective. This regularization term is the KL divergence with the prior distribution. While the prior distribution could be said to have a regularization effect on the posterior, saying that the prior distribution is a regularizer is reductive and probably misleading.


----------------------------------------------

Following the author's rebuttal I think the paper has benefitted from further experiments and from further clarifications. I would like to thank the authors for carefully considering my feedback and for modifying their paper in the directions I suggested. Ultimately, like I said in my original review, I think this is a very interesting and well-motivated problem, but I still have a few doubts. In particular, the doubt about the paper's use of the term of "conjugate" remains. In their rebuttal the authors use the term approximate-conjugate prior, but I am not sure that this is satisfactory as being conjugate means you have knowledge of the form of the true posterior's closed form, which is not the case for BNNs.

I have increased my score to reflect that I think the authors are moving in a promising direction and I hope that they will continue with this work. One thing I will note on the experimental side of things is that having greater variance is indeed interesting, but it may or may not be correlated with increased uncertainty and this may be interesting to investigate in a future version of this work.

---

> ### Author Response · Authors · 2020-11-20
> **Responses for AnonReviewer4**
>
> We really appreciate for your helpful comments and critiques on terminology. After receiving your comments, we thoroughly read the entire paper to address the enhanced clarity and readability. Thanks to your helpful comments, we believe that you will find the revised version of our paper superior to the last in the sense of using proper mathematical terminology.
> ***
>
> **Q1. Misnomers with “true posterior”, “conjugate prior”, and “regularization”.**
>
> A1. To clearly deliver mathematical terms without confusion, we modified the misnomers in terms of “true posterior”, “conjugate prior”, and “regularization”. Updated version to your responses has been uploaded. We mostly modified the expressions of “find/obtain true posterior” to those of “approximate true posterior”. Furthermore, we denoted $q(w)$ as an approximate posterior to approximate the true posterior, and modified the previous expression of $q(\mu, \sigma^2)$ to denote an approximate conjugate prior to approximate the conjugate prior as a same distribution family of the true posterior. Lastly, we modified the expression of “a regularization term for the Gaussian parameters” in page 4 to that of “a regularization term for an approximate posterior”. Besides, other modifications of mathematical terminology have been applied in the updated version of our paper.
>
> ***
>
> **Q2. Robust Optimization is the special case of adversarial training where only adversarial data is used. The authors conflate adversarial optimization (optimization with respect to an adversarial objective) with robust optimization which has a rich history in optimization prior to its application to deep learning. The end of section 2.1 should have its terminology corrected.**
>
> A2. As you noted, Robust Optimization has a rich history in optimization theory prior to its application to deep learning. Therefore, adversarial training is just the special case of robust optimization as you mentioned. Our criteria for dividing the adversarial defense into three methods were to follow the standard categorization of previous studies. Recent papers provided a comprehensive review of adversarial attacks and defenses [1-4]. They also divided adversarial defense approaches into a number of categories including robust optimization. We contemplate that’s why some misunderstandings seem to have occurred in the process of delivering our contents to readers. Thanks to your helpful comments, we modified and clarified the general description of robust optimization and adversarial training at the end of section 2.1 to prevent further misunderstanding for readers. And, we added more references to justify our statements on the terminology. The following sentences represent the end of section 2.1.
>
> *Robust optimization: it is a well-known paradigm that aims to obtain solutions under bounded feasible regions. Especially, its main interest from an adversarial perspective, is improving the classifier's robustness by changing the learning scheme of deep neural networks.*
>
> ***
> References
>
> * [1] Wong, E., & Kolter, Z. (2018, July). Provable defenses against adversarial examples via the convex outer adversarial polytope. In International Conference on Machine Learning (pp. 5286-5295). PMLR.
> * [2] Dong, Y., Fu, Q. A., Yang, X., Pang, T., Su, H., Xiao, Z., & Zhu, J. (2020). Benchmarking Adversarial Robustness on Image Classification. In Proceedings of the IEEE/CVF Conference on Computer Vision and Pattern Recognition (pp. 321-331).
> * [3] Hao Chen, H. X. Y. M., Deb, L. D., Anil, H. L. J. L. T., & Jain, K. (2020). Adversarial attacks and defenses in images, graphs and text: A review. International Journal of Automation and Computing, 17(2), 151-178.
> * [4] Silva, S. H., & Najafirad, P. (2020). Opportunities and challenges in deep learning adversarial robustness: A survey. arXiv preprint arXiv:2007.00753.

---

### Official Review · AnonReviewer3 · 2020-10-30
**Improving Adversarial-BNN with hierarchical variational inference**

**Rating:** 6
**Confidence:** 4

**Review:**

Updates:
The author addressed my concerns about the experiments. Though the improvement is marginal and I still have some concerns, I’m ok to accept the paper. I’ll change my score to 6.
========================

Summary:
The paper studied the adversarial Bayesian Neural Network and found that the stochasticity of it vanished. As stochasticity can help improve robustness against adversarial examples, the author proposed to use conjugate prior of Gaussian posterior to improve stochasticity of the model and robustness at the same time.

Strength:
Experiments show that the proposed method outperforms adversarial-BNN and adversarial training on several benchmark datasets and the stochasticity of the proposed model is larger than adversarial-BNN.

Weakness:
The evaluation is somehow questionable. Checked the original paper of adversarial-BNN and found that the performances of adversarial-BNN is much better than reported in this paper. In both papers, VGG16 is the base structure of BNN, but the reported performances of adv-BNN and adversarial training are different in two papers on CIFAR10.

CIFAR10 Results

| $\epsilon$ | adv-BNN | adv-BNN(in this paper) | adv | adv (in this paper)|
|:------:|:-----:|:------:|:-----:|:------:|
| 0 | 79.7 | 62 | 80.3| 72 |
|0.015 | 68.7 | 54| 58.3 | 60 |

It could be a problem of hyper-parameter tuning. Could the author provide some explanation on this?

In experiments, the models on CIFAR10 and STL10 are trained with $L_\infty$ perturbation magnitude of 0.03. They are evaluated under PGD and EOT-PGD with $L_\infty$ in $[0,0.03]$. The range of attack perturbation magnitude on CIFAR10 and STL10 could be larger, such as $[0,0.08]$, to better compare the baselines with the proposed method.

Clarity and Correctness:
The paper is well written and easy to follow but the experiments might be problematic.

Reproducibility:
Code of the method is not available.

Conclusion:
The idea is clear and novel but experiment results need more elaboration. Overall, I think the paper is marginally below the acceptance threshold. I like the idea of using conjugate prior to improve stochasticity and robustness. However, I'm a little bit concerned about the experiment results. If that can be addressed, I'm willing to accept the paper.

---

> ### Author Response · Authors · 2020-11-20
> **Responses for AnonReviewer3**
>
> We are thankful for your detailed and discerning assessment of our work. As with AnonReviewer2, your major concern is also the difference between the reported performances of the adv-BNN and adversarial training in the original paper and ours.
>
> ***
>
> **Q1. The evaluation is somehow questionable. Checked the original paper of adversarial-BNN and found that the performances of adversarial-BNN is much better than reported in this paper. In both papers, VGG16 is the base structure of BNN, but the reported performances of adv-BNN and adversarial training are different in two papers on CIFAR10. It could be a problem of hyper-parameter tuning.**
>
> A1-1. The issue of different reported performances of adv-BNN
>
> For this issue, you can refer to ["A1."](https://openreview.net/forum?id=Cue2ZEBf12&noteId=yZuA3QFEXSj) in Responses for AnonReviewer2.
>
> A1-2. The issue of different reported performances of normal adversarial training
>
> For the performance of normal adversarial training you pointed out, we used a different experiment setting for optimizer. We used an Adam optimizer for all methods whereas the original source code used an SGD optimizer only for normal adversarial training ([https://github.com/xuanqing94/BayesianDefense/blob/master/main_adv.py#L143](https://github.com/xuanqing94/BayesianDefense/blob/master/main_adv.py#L143)). The original source code used different optimizers for adversarial training(SGD) and adv-BNN(Adam) ([https://github.com/xuanqing94/BayesianDefense/blob/master/main_adv_vi.py#L150](https://github.com/xuanqing94/BayesianDefense/blob/master/main_adv_vi.py#L150)). When we did not change anything in the open source code of the original adv-BNN, the original source code actually used Adam optimizer for adv-BNN in the training process. However, they used an SGD optimizer for the normal adversarial training. If we use an SGD optimizer for adv-BNN in the original source code, then the training becomes unstable and fails to implement with a technical issue. In our study, we used an Adam optimizer for all methods (adv, adv-BNN, and adv-Ours) in order to achieve coherent experimental settings. Therefore, performance differences of adversarial training arise from that difference.
>
> In short, the illustrated Figure 2-4 (also even Figure 3 newly added in updated paper for ablation study) were fairly compared figures with adv, adv-BNN, and adv-Ours on the equal evaluation settings where we utilized the whole layer summation of KL divergence for full Bayesian (for the first issue), and equally employed Adam optimizer (for the second issue).
>
> ***
>
> **Q2. In experiments, the models on CIFAR10 and STL10 are trained with $L_{\infty}$ perturbation magnitude of 0.03. They are evaluated under PGD and EOT-PGD with $L_{\infty}$ in [0,0.03]. The range of attack perturbation magnitude on CIFAR10 and STL10 could be larger, such as [0,0.08], to better compare the baselines with the proposed method.**
>
> A2. We added ablation study for attacking with higher perturbation magnitude in page 8-9. As you noted, the range of attack perturbation magnitude on CIFAR10 and STL10 could be larger. We conducted additional experiments on this larger perturbation range [0, 0.08]. The results of PGD attack are summarized as the following Tables. These results demonstrated that the proposed method outperforms the baseline models as well. The detailed performance against EOT-PGD attack was added in Figure 3 of newly uploaded paper.
>
> * Large range PGD attack on VGG16 trained with $\delta = 8/255$ for CIFAR10
> | $\delta$     	|   0   	| 0.018 	| 0.031 	| 0.043 	| 0.062 	|  0.08 	|
> |:--------:	|:-----:	|:-----:	|:-----:	|:-----:	|:-----:	|:-----:	|
> |    adv   	| 71.37 	| 57.17 	| 45.57 	| 34.26 	| 18.85 	|  8.98 	|
> |  adv-BNN 	| 65.17 	| 54.51 	| 46.80 	| 38.30 	| 26.31 	| 15.58 	|
> | adv-Ours 	| 74.00 	| 61.21 	| 51.56 	| 41.65 	| 27.48 	| 15.89 	|
>
> * Large range PGD attack on Model A trained with $\delta = 8/255$ for STL10
> |     $\delta$ 	|   0   	| 0.018 	| 0.031 	| 0.043 	| 0.062 	| 0.08 	|
> |:--------:	|:-----:	|:-----:	|:-----:	|:-----:	|:-----:	|:----:	|
> |    adv   	| 60.08 	| 32.95 	| 20.37 	| 13.22 	|  6.09 	|  2.8 	|
> |  adv-BNN 	| 62.23 	| 38.27 	| 26.69 	| 18.29 	|  9.06 	| 4.17 	|
> | adv-Ours 	| 61.86 	| 43.84 	| 32.85 	| 22.07 	| 12.23 	| 6.69 	|

---

### Author Response · Authors · 2020-11-20
**Paper update overview**

We would like to show our sincere gratitude for the benefits that we derived from all reviewers. We took all the comments very seriously and regarded them as great opportunities to enhance the overall quality of our paper. We tried to address all of concerns and revised our paper accordingly. We will also answer directly to each review comment. We summarized the most notable changes in our paper:

* We fixed all misnomers and correspondingly clarified the use of each terminology to avoid any misunderstanding throughout the entire paper.
* We added additional ablation studies to address reviewers’ concerns and make elaborate experiments in the modified paper.

We sincerely hope that this revision effort meets your expectations. We are happy to provide further clarification if any of the reviewers’ concerns are not answered.

---

### Decision · Program_Chairs · 2021-01-07
**Final Decision**

**Decision:**

Reject

**Comment:**

This paper improves on previous work (adv-BNN) with hierarchical variational inference. It observes that mean-field VI training for BNNs often result in close-to-deterministic approximate posterior distributions for weights, which effectively makes the BNN closer to deterministic neural network, thereby loosing the robustness advantage of stochastic neural networks. To address this, a hierarchical prior is proposed on the weights, which, together with the corresponding approximate posterior design, aims at preventing the collapse of the variances of the weights towards zero. This improved version of adv-BNN is shown to be reasonably better than the original adv-BNN and their deterministic counter-part against the PGD and EOT attacks on a various of benchmark dataset in the adversarial robustness literature.

Reviewers initially had questions about whether the comparison is fair to the original adv-BNN since the reported results were very different. This issue has been addressed by the authors during the author feedback period, after that reviewers agreed that the proposed approach is a good extension of adv-BNN towards making it more robust. They also agree that the analysis of the original adv-BNN in terms of posterior variance collapse is interesting and potentially useful, although they also pointed out the link of increased variance (with the proposed method) and better uncertainty estimation is unclear.

In revision, I would encourage the authors to clear up the confusions of the reviewers by clearly stating the comparison setting with the original adv-BNN, and better clarify the methodology.